# Synthesis of L-Lactide from Lactic Acid and Production of PLA Pellets: Full-Cycle Laboratory-Scale Technology

**DOI:** 10.3390/polym16050624

**Published:** 2024-02-25

**Authors:** Gadir Aliev, Roman Toms, Pavel Melnikov, Alexander Gervald, Leonid Glushchenko, Nikita Sedush, Sergei Chvalun

**Affiliations:** 1M.V. Lomonosov Institute of Fine Chemical Technologies, MIREA—Russian Technological University, Moscow 119571, Russia; zaligad99@gmail.com (G.A.); toms.roman@gmail.com (R.T.); gervald@bk.ru (A.G.); 2InTechnoBioMed, Ulyanovsk 432072, Russia; ileo.glu@gmail.com; 3Enikolopov Institute of Synthetic Polymeric Materials, Russian Academy of Sciences, Moscow 117393, Russia; nsedush@gmail.com (N.S.); s-chvalun@yandex.ru (S.C.)

**Keywords:** lactide, polylactide, lactic acid, tin octoate, ring-opening polymerization, lactic acid oligomerization, PLA pellets

## Abstract

Lactide is one of the most popular and promising monomers for the synthesis of biocompatible and biodegradable polylactide and its copolymers. The goal of this work was to carry out a full cycle of polylactide production from lactic acid. Process conditions and ratios of reagents were optimized, and the key properties of the synthesized polymers were investigated. The influence of synthesis conditions and the molecular weight of lactic acid oligomers on the yield of lactide was studied. Lactide polymerization was first carried out in a 500 mL flask and then scaled up and carried out in a 2000 mL laboratory reactor setup with a combined extruder. Initially, the lactic acid solution was concentrated to remove free water; then, the oligomerization and synthesis of lactide were carried out in one flask in the presence of various concentrations of tin octoate catalyst at temperatures from 150 to 210 °C. The yield of lactide was 67–69%. The resulting raw lactide was purified by recrystallization in solvents. The yield of lactide after recrystallization in butyl acetate (selected as the optimal solvent for laboratory purification) was 41.4%. Further, the polymerization of lactide was carried out in a reactor unit at a tin octoate catalyst concentration of 500 ppm. Conversion was 95%; Mw = 228 kDa; and PDI = 1.94. The resulting products were studied by differential scanning calorimetry, NMR spectroscopy and gel permeation chromatography. The resulting polylactide in the form of pellets was obtained using an extruder and a pelletizer.

## 1. Introduction

Lactide is a monomer for the synthesis of biocompatible polylactide (PLA) and copolymers based on it. PLA is a biodegradable polymer that does not cause negative effects and pathological changes when used in living organisms. It is used to produce tissue-engineered scaffolds, drug delivery systems or coating membranes, and various bioabsorbable medical implants, and it is also worth mentioning the use of this material in dermatology and cosmetology [1,2,3]. L-isomers of lactide are used in the synthesis of polymers for further use in medicine for the production of orthopedic implants, such as screws, plates and pins, as well as other implants that should maintain high strength for a relatively long period of 6 months and more [4]. Poly(L-lactide) is also widely used for the production of biodegradable packaging [4]. The stereoisomers of lactide are shown in Figure 1. The copolymerization of different stereoisomers is an effective way to control the properties and degradation profile of PLA. It is worth noting that PLA is singled out among biopolymers because its main valuable qualities are considered to be environmental friendliness, biocompatibility and recyclability; it is non-toxic, as are its degradation products [5,6]. Biodegradability is one of the key advantages of PLA, distinguishing it from other polymers, the disposal of which requires mechanical (physical) [7] or chemical recycling [8,9] or incineration.

There are various ways to produce lactide, but the most promising one is considered to be depolymerization from lactic acid oligomers. Lactic acid is naturally produced by a wide range of microorganisms, including bacteria, yeasts and filamentous fungi; hence, its production by enzymatic fermentation is common in industrial production [10,11]. Thus, the bulk of lactic acid is produced by microbial fermentation of all types of sugar sources, ranging from pure monosaccharides found in beets to more complex polysaccharides from molasses or whey. The most common substrates for lactic acid bacteria are glucose and sucrose [6,7]. High production efficiency is achieved as a result of the fact that some subspecies of bacteria are able to produce two molecules of lactide from one molecule of monosaccharide. Lactic acid exists as two optically active stereoisomers: L(+) and D(–) forms. Therefore, the form determines what kind of product will be obtained. To synthesize pure L-lactide, it is required to use L-lactic acid and sustain the reaction conditions that minimize the production of meso-lactide. Namely, it is the necessary to maintain the temperature and pressure regime and use catalysts that allow the necessary stereospecificity to be achieved [8,9]. Currently, a two-step method consisting of lactic acid oligomerization followed by depolymerization to form lactide is commonly used. However, the production of lactide requires high temperature, vacuum and energy input, and the subsequent ring polymerization requires high-purity lactide to produce PLA. Studies on this process have shown that the purity of lactide has a significant effect on the polymerization process and, consequently, on the molecular weight of the resulting polymer [12,13]. Therefore, it is necessary to purify lactide to obtain PLA with the required physical and mechanical properties and high molecular weight of up to 500 kDa [14]. Thus, the production of high-molecular-weight PLA is a complex multistage process. It is especially important that all technological stages from monomer synthesis to the production of the polymer in a convenient commercial form (pellets) are implemented in one place. This avoids unnecessary technological operations, optimizes the process and reduces its duration. The aim of this work was to implement the complete cycle of high-molecular-weight PLA production from lactic acid in a single installation: synthesis of lactide, its purification, and the production and processing of PLA. Studies were carried out on the resulting intermediate products, as well as the influence of certain factors on their properties in each stage.

## 2. Materials and Methods

### 2.1. Materials

An aqueous solution of L-lactic acid (80%) with stereochemical purity of 95% (Henan Jindan lactic acid Technology Co., Ltd., Henan, China) was used. The catalyst tin octoate (Dabco T-9; Evonik Industries AG, Staufen, Germany) was used without further purification. Ethanol, isopropanol, ethyl acetate, butyl acetate and chloroform (>99%) were purchased from Ekos-1 (Staraya Kupavna, Russia) and were used without further purification. L-lactide of medical grade (Corbion, Amsterdam, The Netherlands) was used as a comparison sample.

### 2.2. Lactic Acid Concentration

Under laboratory conditions, it is more efficient to perform lactic acid concentration in a rotary evaporator. The free-water removal process was carried out in a rotary evaporator, IKA RV 10 (IKA-Werke GmbH & Co. KG, Staufen, Germany) (shown in Figure 2), with pressure reduction to 50 mbar and constant stirring at 100 rpm; the temperature of the water bath was raised stepwise to 90 °C. The lactic acid concentration process was carried out for two hours and was controlled by the gravimetric method.

### 2.3. Oligomerization of Lactic Acid

The polycondensation of lactic acid was carried out in a 500 mL three-neck flask equipped with a magnetic stirrer and connected to a receiving flask equipped with a direct refrigerator. A total of 250 g of concentrated lactic acid was poured in the flask, and the catalyst tin octoate was added in an amount of 0.25%, 0.50% or 1.00% by weight of lactic acid, depending on the synthesis option. The flask was purged with nitrogen (99.96% purity) during the process, and the temperature setting from 140 °C to 200 °C was maintained by a flask heater, Joanlab 22403-HMC (JOAN LAB Equipment Co., Ltd., Huzhou, China), with a measurement accuracy of ±1 °C.

The polycondensation process was carried out at gradually decreasing pressure in the range from 510 to 10 mbar for 3 h. Samples were taken at certain time points to characterize the molecular weight; the mass and refractive index of the evaporated water were used to determine the conversion stage.

### 2.4. Synthesis of Lactide

Synthesis was carried out on the same setup (Figure 3). When the release of water from the reaction flask stopped, the receiving flask was replaced, and the temperature was increased to 210 °C. The pressure in the reaction system was maintained at 5–10 mbar for one and a half hours. The raw lactide was isolated under vacuum into a receiving flask immersed in a water bath at a temperature of 10 °C [15,16,17,18,19].

The scaled-up synthesis was carried out in a 2 L flask with a total loading mass of 1000 g under similar conditions and formulation to obtain a sufficient volume of lactide to be processed. The process was carried out for 6 h. Conversion and yield were determined by the gravimetric method according to the formulas below.

Raw lactide yield:(1)$$K_{\mathrm{raw}-\mathrm{La}}=m_{\mathrm{raw}-\mathrm{La}}/m_{\mathrm{conc}-\mathrm{La}}$$

Practical lactide yield:(2)$$K_{p}=m_{\mathrm{pl}}/m_{\mathrm{conc}-\mathrm{La}}$$

Specific yield of lactide:(3)$$K_{t}=m_{l}/m_{\mathrm{la}}$$

Yield after recrystallization:(4)$$K_{r}=m_{\mathrm{pl}(i)}/m_{\mathrm{pl}(i-1)}$$

Polymerization conversion of lactide:(5)$$K_{\mathrm{pla}}=m_{\mathrm{pla}}/m_{\mathrm{pl}}$$
where the following apply:

m_raw-La_—mass of raw lactide (g);

m_conc-La_—mass of concentrated lactic acid (g);

m_pl_—weight of purified lactide (g);

m_l_—mass of purified lactide (g);

m_la_—mass of lactic acid (g);

m_pl(i)_—mass of lactide after the i stage of purification (g);

m_pl(i–1)_—mass of lactide after (i – 1) stage of purification (g);

m_pla_—mass of polylactide (g).

### 2.5. Purification of Raw Lactide

The purification of raw lactide from residual oligomers, lactic acid and water was carried out by recrystallization from ethanol, isopropanol, ethyl acetate and butyl acetate. The process was carried out in a 100 mL beaker on a heating plate equipped with a magnetic stirrer. The ratio of raw lactide:solvent in the case of alcohols and ethyl acetate was taken in the mass ratio of 2:1, and in case of butyl acetate—3:1. The temperature of 70 °C and stirring were maintained by a heating plate, IKA C-MAG HS-7 (IKA-Werke GmbH & Co., KG, Staufen, Germany). After complete dissolution of the substance, the beaker was removed from the hotplate and cooled to 5 °C. Then, lactide crystals were separated from the solvent by using Buchner funnel vacuum filtration.

### 2.6. Synthesis of Polylactide

The ROP of lactide for research purposes was carried out in a 100 mL three-neck flask equipped with a top-driven stirrer (Figure 4). In total, 30 g of purified lactide and 0.042 g (10^–3^ mol/L) of tin octoate were placed in the flask. The flask without moisture was purged with nitrogen, isolated and placed in an oil bath before starting the synthesis. The temperature setting from 160 °C to 190 °C was maintained by a heating plate, IKA C-MAG HS-7. Samples were taken during the reaction to determine the molecular weight characteristics. The reaction mixture was diluted with chloroform and precipitated in methanol after the completion of polymerization. Then, PLA was dried at 60 °C to constant weight in a vacuum-drying oven. The polymer was processed in an extruder, followed by cooling of the filament and its granulation to obtain PLA pellets.

### 2.7. Differential Scanning Calorimetry (DSC)

The thermal effect of lactide and PLA melting was studied with a differential scanning calorimeter, Netzsch DSC 204 (NETZSCH GmbH & Co. Holding KG, Selb, Germany), in a dry inert atmosphere (argon 99.99%) at a flow rate of 20 mL/min in the range from 10 to 260 °C with a heating rate of 10 °C/min. A prepared sample weighing from 4 to 7 mg was taken and placed in a standard aluminum crucible with a lid. The results were processed using the Netzsch Proteus program. The software was used to determine the melting onset temperature, extremum temperature and thermal effect (area under the peak).

### 2.8. Gel Permeation Chromatography (GPC)

Molecular weight characteristics were investigated by gel permeation chromatography (Gilson Inc., Villiers-le-Bel, France). The analysis was carried out at 25 °C in tetrahydrofuran (THF) at a flow rate of 1.0 mL/min with a refractive index detector. For the separation and identification of oligomeric fractions, a PLgel 3 µm MIXED E column (separating capacity of up to 30 kDa) (Agilent, Santa Clara, CA, USA) was used, with a set of polystyrene standards with Mp (peak molecular weight) of 580, 1280, 2940, 10,110 and 28,770 g/mol and PDI lower than 1.12 (Agilent). For the separation and identification of high-molecular-weight polymers, a PLgel 5 µm MIXED B column (separating capacity of 500–3,000,000 Da) (Agilent, Santa Clara, CA, USA) was used, with a set of polystyrene standards with Mp (peak molecular weight) of 2940, 10,110, 28,770, 10,110, 74,800, 230,900 and 1,390,000 g/mol and PDI lower than 1.12 (Agilent). A solution of polymer in eluent was prepared for analysis with polymer concentration not exceeding 5 mg/mL and not less than 1 mg/mL.

### 2.9. Nuclear Magnetic Resonance (NMR)

The qualitative and quantitative composition of lactic acid, lactide and PLA was studied by using an NMR spectrometer (1H NMR), BrukerDPX-500 (Bruker Corporation, Bremen, Germany), with Fourier transform in CDCl_3_ (internal standard: TMS).

### 2.10. Determination of Refractive Index

The refractive index was determined using the BS RMF 960 refractometer (Bellingham and Stanley Ltd., Tunbridge Wells, United Kingdom).

## 3. Results and Discussion

### 3.1. Lactic Acid Concentration

The refractive index of the distilled liquid was determined during each of the free-water distillation processes. The measured values ranged from 1.3451 to 1.3457, which are not significantly different from the refractive index of water (1.3333). Thus, most of the lactic acid remained in the reaction system and did not enter the distillation flask in the initial stages of synthesis. Aqueous solutions of lactic acid with a given concentration were also prepared; the refractive indices of 5% and 80% aqueous solutions of lactic acid were 1.3404 and 1.440, respectively. From the data obtained, it can be determined that the concentration of lactic acid in the removed water during the concentration process was approximately 9.0–9.2%. The mass of the extracted fraction was typically 19.7–19.8% of the mass of the loaded aqueous lactic acid solution.

Lactic acid oligomer is a yellowish-orange, slightly viscous liquid after concentration. The analysis of the concentrated lactic acid and initial solution by the GPC method showed the presence of not only lactic acid but also di-, tri- and tetramers in the system, which was also confirmed by NMR. It was found that the average molecular weight of the components increased after water evaporation. Figure 5 and Figure 6 show the molecular weight distribution curve and 1H NMR spectrum of concentrated lactic acid, respectively. The initial lactic acid solution contained approximately 25% oligomers (di- and trimers) by weight. The content of oligomers (di-, tri- and tetramers) after water evaporation was already 45–50% by weight. Thus, we can call this process pre-oligomerization. The NMR spectrum (Figure 6) shows a peak at 1.46 ppm, which corresponds to the protons of the -CH_3_ group (number 2) close to the -OH in lactic acid oligomers. The peak at 4.37 ppm corresponds to both the protons of the -CH group (number 1) in lactic acid oligomers. There are also peaks at 1.56 and 5.17 ppm, which correspond to protons in the -CH_3_ (number 4) and -CH (number 3) groups of the oligomers. The ratio of their areas (numbers 4 and 3) is 3.04/1 which is very close to the theoretical 3/1.

### 3.2. Synthesis of Lactide and Its Characterization

The scheme of lactide production can be briefly represented as a two-step process: oligomerization of lactic acid and its depolymerization with cyclic dimer (lactide) detachment. The synthesis was carried out at different concentrations of tin octoate catalyst: 0.25%, 0.50% and 1.00% by weight of lactic acid. Tin octoate was chosen as the optimal catalyst for this reaction based on the literature data because it has stereoselective properties that would allow to purer L-lactide to be obtained while minimizing the formation of D-form and meso-lactide [20,21,22]. It is worth noting that this type of catalyst has been approved by the US Food and Drug Administration (FDA) for use as an additive suitable for food contact and as a polymer packaging [23]. Tin octoate was injected immediately to achieve the greatest efficiency. A schematic representation of the reaction for the production of lactic acid oligomers in the presence of tin octoate and their depolymerization to lactide is shown in Figure 7. The weight average molecular weights in relation to the conversion steps (or time) of syntheses No. 1–3 are presented in Table 1, and the GPC curves are presented in Figure 8, Figure 9 and Figure 10. Figure 11 shows the dependence of oligomer molecular weight on the mass fraction of stripped reaction water in different stages of conversion. It was observed that all three syntheses proceeded the same way, as the curves converge into one graphical dependence.

The main distinguishing parameter during the syntheses was the amount of catalyst introduced. It was observed that when the concentration of the introduced catalyst was increased up to 1.00 wt%, the weight average molecular weight of the oligomers was higher compared with the values obtained when a lower amount of catalyst was introduced. The molecular weights (MWs) of the oligomers before the start of the distillation of lactide were 860, 1670 and 1995 Da, respectively. The refractive index values of the resulting distillate were 1.3457, 1.3435 and 1.3427, respectively. Thus, an increase in the amount of catalyst led to a decrease in the amount of lactic acid in the distillate under the same temperature and pressure conditions. The first stage of lactide production (oligomerization of lactic acid) is followed by the stage of depolymerization of oligomers with lactide detachment. This stage is carried out at increasing temperature and decreasing pressure [24,25,26]. The values of raw lactide yields based on the mass of concentrated lactic acid are presented in Table 1. The raw lactide was analyzed using NMR after the completion of the syntheses. The spectra are presented in Figure 12.

The 1H-NMR spectrum of raw lactide exhibits a signal (number 2′) which corresponds to the methyl protons of meso-lactide and a signal (number 2) which relates to the methyl protons of L-lactide or D-lactide. Signal number 5 shows the presence of methyl group protons in the oligomers of PLA. Signals number 3 and 4 are associated with the proton vibrations at the carbon in the -CH and -CH_3_ groups of lactic acid oligomers. Signal number 1 is related to the -CH groups in the lactide ring. The content of meso-lactide in raw lactide decreased with the increase in catalyst content (calculated as the ration of peaks 2 and 2′).

A formulation with a tin octoate concentration of 0.50 wt.% was chosen as optimal to obtain more raw lactide for recrystallization and further synthesis of PLA, since it allowed us to carry out the synthesis at a sufficiently high rate and low yield of meso-lactide (since the percentage of meso-lactide increases with the increase in catalyst concentration). An additional reduction in meso-lactide content was achieved by optimizing the synthesis temperature regime.

The samples were also taken during the synthesis process for molecular weight determination. It was found that the molecular weight growth does not stop at the oligomer synthesis stage (Figure 13). This increases the viscosity of the system, which further reduces the yield of lactide. Figure 14 shows the plot of the time dependence of the exhausted fraction. The total mass of reaction water before increasing the temperature and decreasing the pressure in the system and the first precipitation of lactide crystals was 15.6% (Figure 14, black line). The mass of raw lactide was weighed after the release of lactide crystals began (Figure 14, red line).

A black-color product with high viscosity formed in the bottom part at the end of the synthesis. This product is the main problem of low lactide yield. This product is hard, brittle and slightly soluble in solvents at room temperature. The scaling process was carried out with a yield of 48%, which made it possible to obtain 487 g of raw lactide from 1 kg of concentrated lactic acid. A lower yield of lactide (48% instead of 68%) also made it possible to reduce the meso-lactide content in raw lactide. The obtained raw lactide was further purified by recrystallization from solvents.

### 3.3. Purification of Raw Lactide and Its Characterization

Properties such as molecular weight, melting point, degree of crystallinity and mechanical strength of the resulting polymer improve as the degree of purification of lactide increases [27]. Thus, increased meso-lactide content in the monomer leads to a decrease in the polymer’s resistance to elevated temperatures during transportation, storage and use of the material. Such PLA, despite the ease of processing, is only suitable as an amorphous material for packaging production. Meso-lactide is an unavoidable by-product in the synthesis of L- and D-isomers, which must be separated in the purification step to increase stoichiometric purity. It was found that the presence of 1% meso-lactide in the reaction mixture during the copolymerization of L- and meso-forms causes a decrease in the melting point of the resulting copolymer by 3 °C. Crystallization occurs more than twice as slow as in PLLA under the same conditions when this concentration increases to 3% [27]. PLA grades for applications requiring better thermal stability are achieved by stereocomplexation with PDLA, which is only effective if raw materials of high stereochemical purity are used [28]. In the stage of lactide synthesis, lactic acid, its oligomers, water, meso-lactide and other impurities, in addition to the product itself, enter the receiving vessel, depending on the reagents used. Five main separation methods are used to purify raw lactide. The most frequently used laboratory method of lactide purification is recrystallization from solvents. This method could give the highest-purity product (purity above 99.5%) in four–five iterations, but at the same time, a large volume of the desired component of the mixture is lost (up to 80% in the case of ethyl acetate) [29]. Most often, lactide is dissolved in alcohols (methanol, ethanol and isopropanol), and oxygenated (ethyl acetate, butyl acetate, acetone, butanone and tetrahydrofuran) and organochlorine (methylene chloride and chloroform) solvents, as well as in organic solvents (benzene, toluene and xylene). Lactide is hydrolyzed in water at room temperature to lactic acid. It should be noted that the rate of hydrolysis of meso-lactide is higher than that of D- and L-lactide. The solubility of lactide in the above-mentioned solvents is proportional to the exposure temperature [30,31]. The most effective solvents for the removal of meso-lactide to be considered should be toluene, isobutanol, ethanol and heptane and their mixtures. Their use can reduce the content of the meso-form by 5 times and of lactic acid by 8.5 times [32,33]. The purification of raw lactide was carried out from acetic acid esters (ethyl and butyl) and alcohols (ethyl and isopropyl). Alcohols dissolve lactic acid oligomers better, while the solubility of lactide is lower than in the case of esters. However, the obtained product is contaminated with hydroxyl groups, which makes it impossible to obtain high-molecular-weight PLA. The yield of the product is higher when alcohols are used as solvents, so their use is reasonable only in the first iterations of recrystallization, and in further steps, it is advisable to use esters to obtain lactide of high purity. Figure 15 shows the results of four iterations of purification of raw lactide from different solvents. It can be seen that isopropyl alcohol gives the highest yield of lactide after recrystallization. The solubility of lactide in this case is lower than when using ethyl acetate; hence, higher product yields are obtained. However, butyl acetate is the most suitable among the used solvents to obtain high-molecular-weight PLA, since it does not contaminate the product with hydroxyl groups. Figure 16 shows the GPC curve of raw lactide before and after the first recrystallization. It can be seen that even the first recrystallization allows the elimination of the main lactide impurities.

The degree of purification of the product after four iterations of recrystallization was determined using the DSC method. The melting point of L-lactide of the highest purification degree was 96 °C, while those of the meso-lactide and racemic mixture of L- and D-lactide were 54 °C and 125 °C, respectively [17].

The degree of purification of a substance can be determined indirectly by the narrowing of the peak and an increase in the degree of its symmetry, since pure substances melt in a narrow temperature range. Figure 17 shows that the majority of lactic acid, oligomers, water and meso-lactide contaminating the L-form remained in the mother liquor after the first recrystallization step. As the number of recrystallization iterations increases, the peak T_melting_ in the graph becomes more intense. Subsequent recrystallization steps do not affect the purity of the substance, but at the same time, they reduce the yield of the product, since part of lactide remains dissolved in the mother liquor. Table 2 shows the values of the initial melting points of lactide obtained with each recrystallization step using all the above-mentioned solvents. It can be seen that four to five recrystallization iterations are sufficient to achieve a high degree of product purification. Figure 18 shows the appearance of lactide before purification by recrystallization and after four recrystallization iterations in butyl acetate.

Figure 19 shows the comparison of the melting temperature of the commercial sample with laboratory lactide after four recrystallization iterations in butyl acetate; their melting thermograms practically coincide.

The composition of lactide was also analyzed by NMR. The 1H-NMR spectra of raw lactide and lactide purified by recrystallization are shown in Figure 20. The 1H NMR spectrum of raw lactide shows signals at 1.6 (indicated by number 2′ in the figure), corresponding to methyl protons in meso-lactide, and signals at 1.5–1.55 (number 2), which are characteristic of methyl protons in L-lactide. The resonance in the range 1.42–1.48 (number 5) shows the presence of methyl group protons in oligomers. The presence of residual monomer is also indicated by resonances in the ranges 1.32–1.38 (number 4) and 4.2–4.4 (number 3), which are responsible for the proton vibrations at the carbon in the -CH and -CH_3_ groups of lactic acid.

It can be seen that after purification by recrystallization, the signals disappeared in the regions, indicating the presence of lactic acid, oligomers and meso-lactide, and the only signals that remained were those in the ranges 5.0–5.10 (number 1) and 1.50–1.55 (number 2), which are responsible for the presence of protons in the -CH and -CH_3_ groups of L-lactide. Thus, the NMR spectrum confirms the purification procedure efficiency and purity of the resulting lactide.

### 3.4. Synthesis of Polylactide and Its Characterization

The molecular weight is the main characteristic of PLA, which determines its further application. It affects its strength, heat resistance, viscosity and decomposition rate. The higher the molecular weight, the higher the strength and heat resistance of the polymer. The analysis of commercial brands of biomedical PLA shows that the optimal molecular weight is considered to be between 50 and 500 kDa [34]. The molecular weight and structure of PLA are regulated by the presence of activator molecules of linear or branched structures containing hydroxyl groups [35,36,37,38,39,40,41,42,43]. The molecular weight of the synthesized PLA will indicate how pure the lactide is (how few hydroxyl-containing components it contains). Undoubtedly, the cleanliness of the reaction vessel and atmosphere is also necessary to obtain high molecular weights. Figure 21 shows a schematic representation of the reaction for the production of PLA by ROP.

In this work, the polymerization of purified lactide was carried out by the mechanism of ROP in the presence of tin octoate as a catalyst at a concentration of 500 ppm at the temperature of 190 °C. The synthesis time was 50 min, and the conversion achieved was 96%. The dependence of conversion on reaction time for PLA production is shown in Figure 22.

The molecular weight characteristics of the obtained polymer were determined by the GPC method. The polymer with a weight average molecular mass equal to 228 kDa was obtained after 50 min of synthesis. The resulting molecular weight of the polymer is sufficient to synthesize copolymers of various molecular weights and structures based on purified lactide. Figure 23 shows the molecular weight distribution during the synthesis of PLA.

Figure 24 represents the DSC curve for the resulting PLA. The graph exhibits a kink at a temperature of 62.5 °C, which corresponds to the glass transition temperature of the obtained PLA. The exothermic peak at 110.5 °C corresponds to the crystallization process of amorphous PLA, and the endothermic peak starting at 172.8 °C corresponds to the polymer melting process. The obtained thermal characteristics correlate well with the literature data [33,44,45,46]. The synthesis conditions were in accordance with the laboratory regulations described above. The residual lactide was removed by vacuum distillation. The synthesized polymer contained a minimal amount of impurities and was subsequently purified by recrystallization in chloroform. The polymer was then processed using a laboratory extruder and granulated to obtain PLA pellets for the convenience of further use in the production of fibers, filament, blending, etc. The obtained pellets had a cylindrical shape with an average size of about 1.5 × 1.5 mm. The obtained intermediate products of the syntheses and pellets are shown in Figure 25.

The resulting molecular weight of the polymer is sufficient to synthesize copolymers of various molecular weights and structures based on purified lactide. Particularly, copolymers of L-lactide with D-lactide and/or glycolide, i.e., PLGA, could be synthesized to tailor the crystallinity and mechanical properties of the materials, as well as their solubility and degradation rate. Such glycolide-enriched copolymers are widely used for fabrication of absorbable surgical sutures, while copolymers of D,L-lactide with glycolide are promising for the development of novel targeted drug nanoformulations and controlled drug release systems based on PLGA microparticles [4,47]. Copolymerization of L-lactide with ϵ-caprolactone is an effective approach to tuning the thermal and mechanical properties of the materials, providing a way to fabricate more soft and elastic materials with increased toughness for various biomedical and technical applications [48].

## 4. Conclusions

We have successfully implemented a full cycle for the production of high-molecular-weight PLA from lactic acid, and the following conclusions could be drawn from the completed work:L-lactide was synthesized by depolymerization of PLA oligomers with the introduction of tin octoate at different concentrations into the system as a catalyst. It was found that the optimal amount of catalyst is about 0.50 wt.%.Several types of solvents were investigated for the recrystallization stage. It was determined that butyl acetate is the most suitable for purifying raw lactide from lactic acid and oligomers. The yield after four-fold recrystallization was 47%, and the presence of lactic acid was not detected. It was also shown, by the DSC method, that after four recrystallization iterations, the melting point of lactide stopped changing and amounted to 97.2 °C. The thermograms also lack peaks not associated with the melting of L-lactide.The data obtained by GPC confirmed that the purity degree of the obtained L-lactide is sufficient for the synthesis of high-molecular-weight PLA, where M_w_ > 200 kDa. The weight average molecular weight of the synthesized lactide was 228 kDa, and the polydispersity index was 1.94. The resulting molecular weight of the polymer is sufficient to synthesize copolymers of various molecular weights and structures based on purified lactide.An effective laboratory installation for the production of PLA pellets with a total loading of lactide from 100 to 400 g was developed. It makes it possible to obtain and test PLA with different molecular weights and compositions. In subsequent works, the purification of lactide will be examined in more detail, and its efficiency will be increased. Moreover, copolymers will also be synthesized in a laboratory installation in order to obtain granules with subsequent processing into materials of various forms.

## Figures and Tables

**Figure 1 polymers-16-00624-f001:**
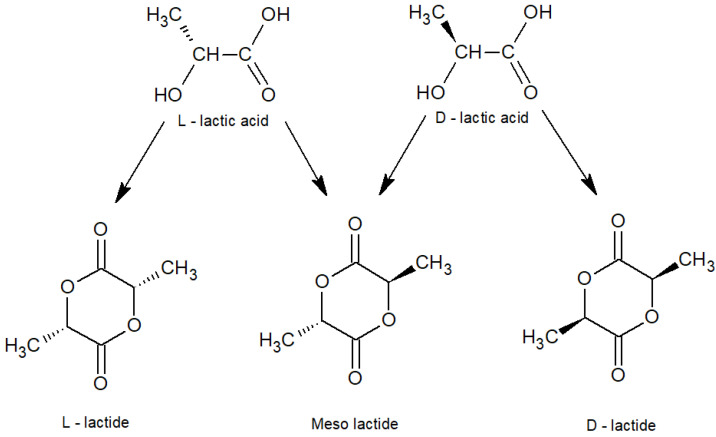
Stereoisomers of lactide.

**Figure 2 polymers-16-00624-f002:**
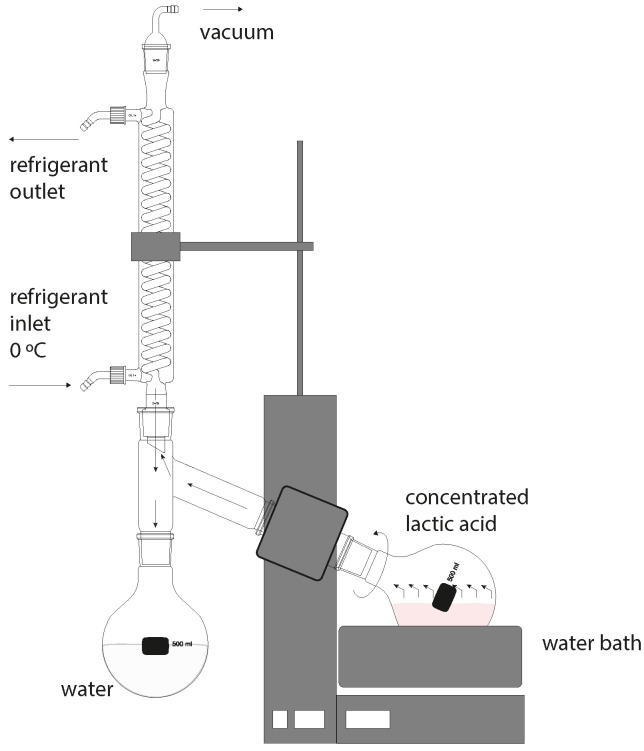
Schematic representation of a vacuum rotary evaporator for lactide concentration.

**Figure 3 polymers-16-00624-f003:**
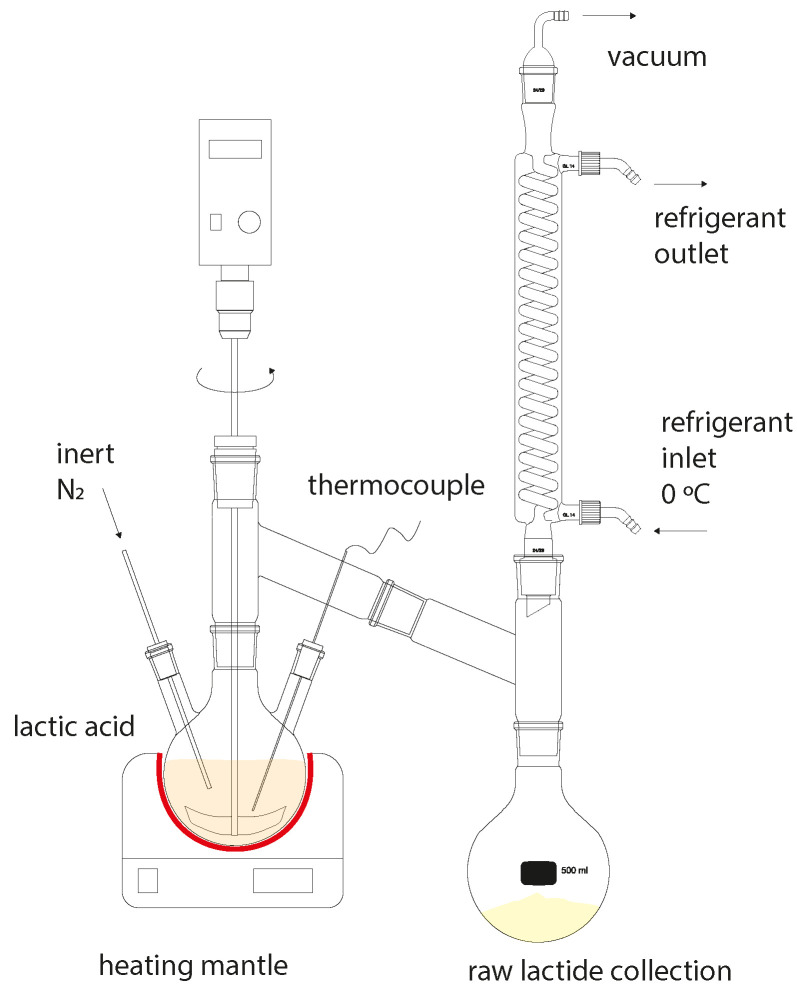
Laboratory setup for lactide production.

**Figure 4 polymers-16-00624-f004:**
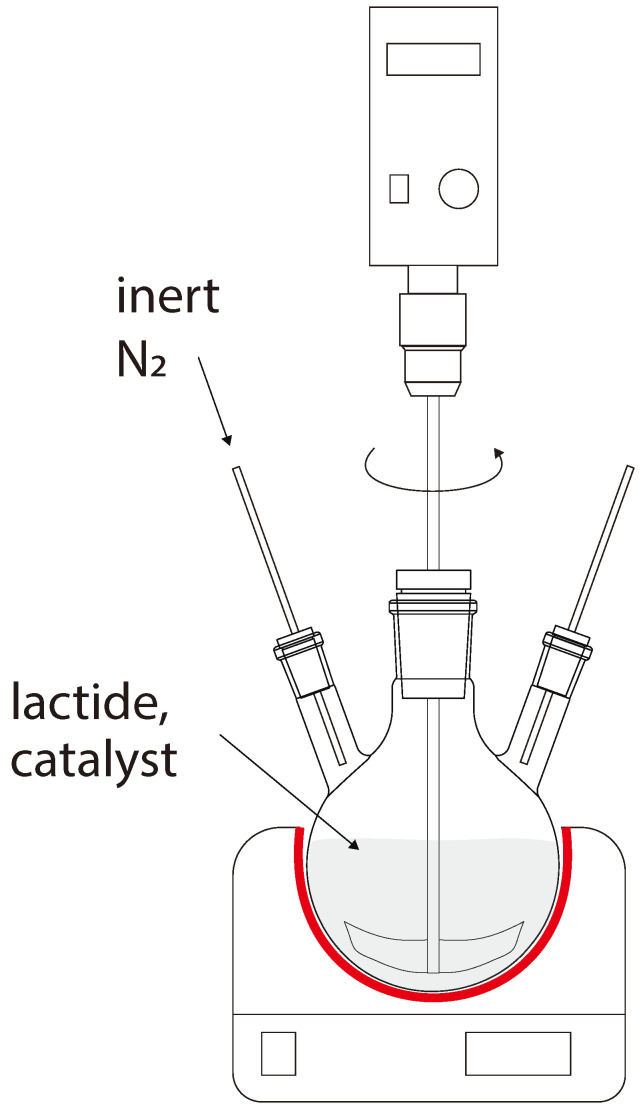
Laboratory setup for PLA production.

**Figure 5 polymers-16-00624-f005:**
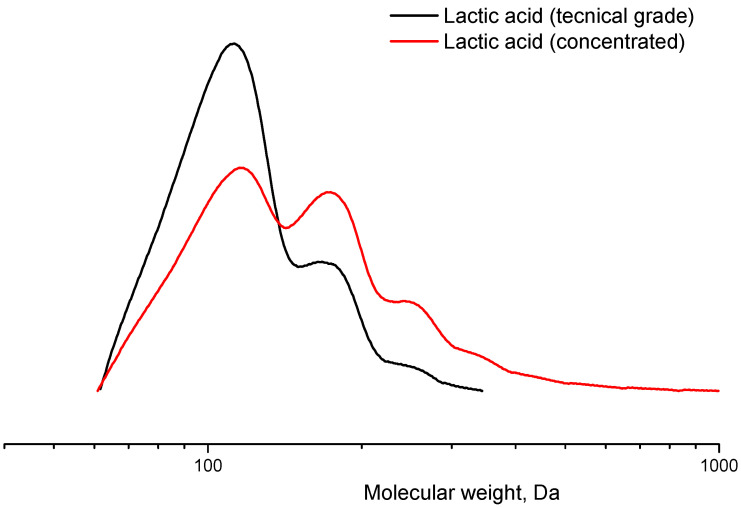
Molecular weight distribution curve of a sample of concentrated lactic acid and lactic acid solution obtained by the GPC method. Temperature: 25 °C. Eluent: tetrahydrofuran (THF) at a flow rate of 1.0 mL/min.

**Figure 6 polymers-16-00624-f006:**
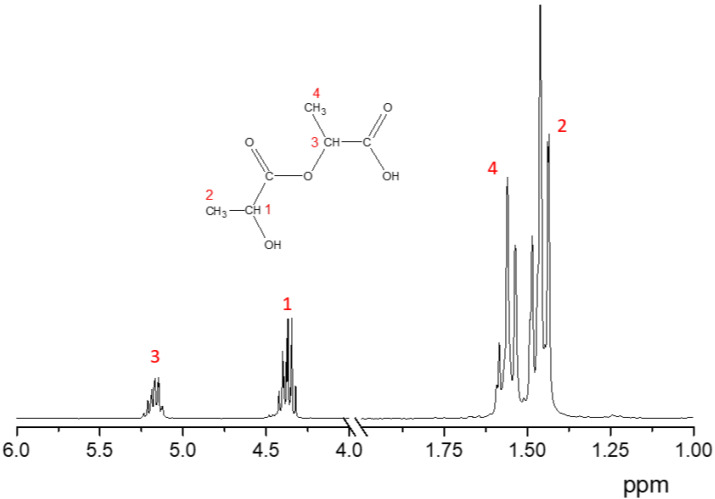
1H NMR spectrum of concentrated lactic acid. Solvent: CDCl_3_ (internal standard—TMS).

**Figure 7 polymers-16-00624-f007:**
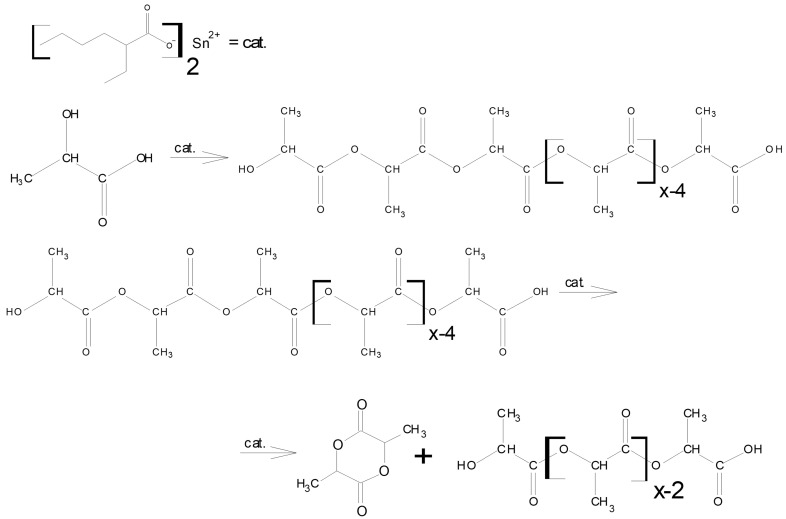
Schematic representation of the preparation of lactic acid oligomers in the presence of tin octoate and their depolymerization into lactide.

**Figure 8 polymers-16-00624-f008:**
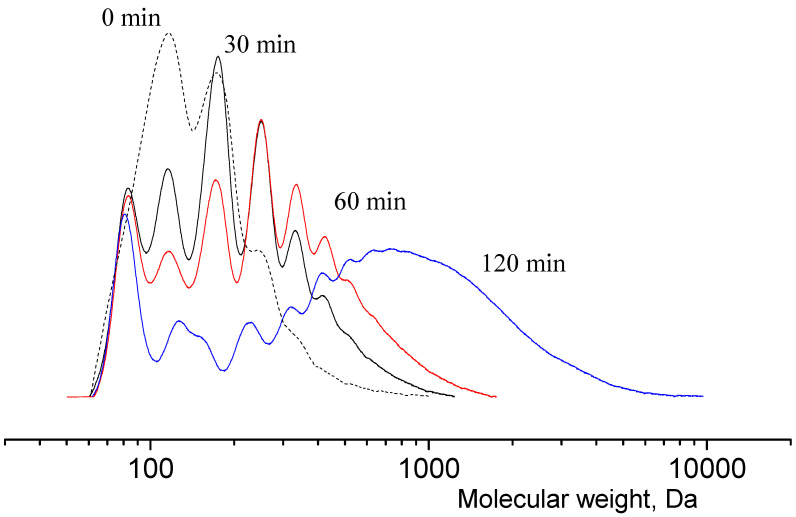
GPC curves of lactic acid oligomers of synthesis 1. Temperature: 25 °C. Eluent: tetrahydrofuran (THF) with a flow rate of 1.0 mL/min.

**Figure 9 polymers-16-00624-f009:**
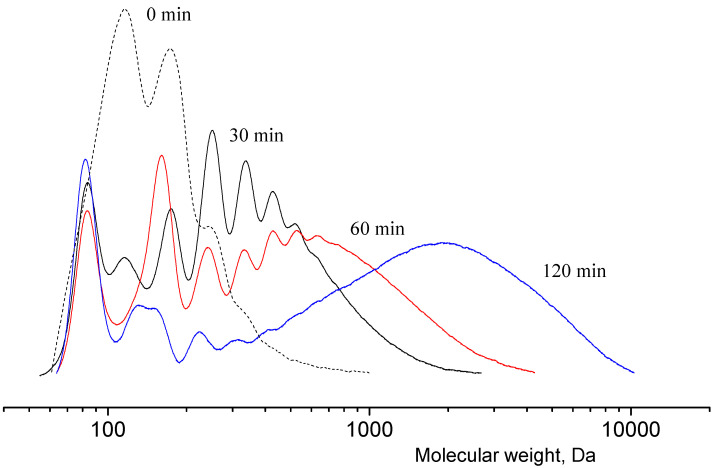
GPC curves of lactic acid oligomers of synthesis 2. Temperature: 25 °C. Eluent: tetrahydrofuran (THF) with a flow rate of 1.0 mL/min.

**Figure 10 polymers-16-00624-f010:**
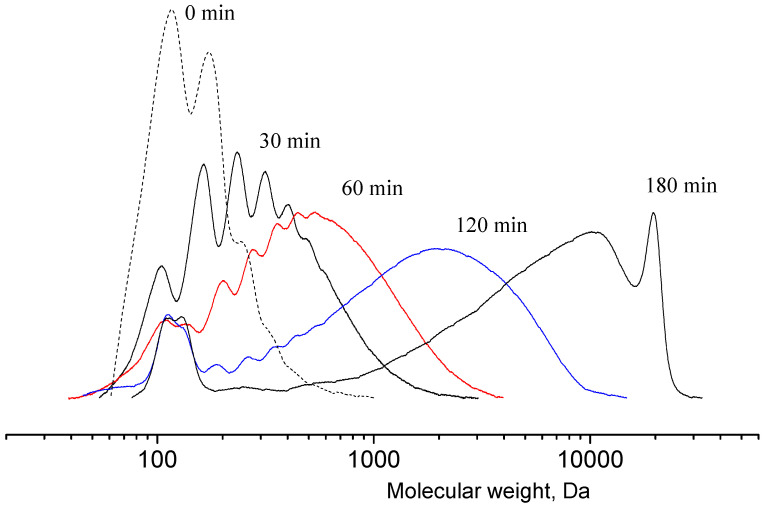
GPC curves of lactic acid oligomers of synthesis 3. Temperature: 25 °C. Eluent: tetrahydrofuran (THF) with a flow rate of 1.0 mL/min.

**Figure 11 polymers-16-00624-f011:**
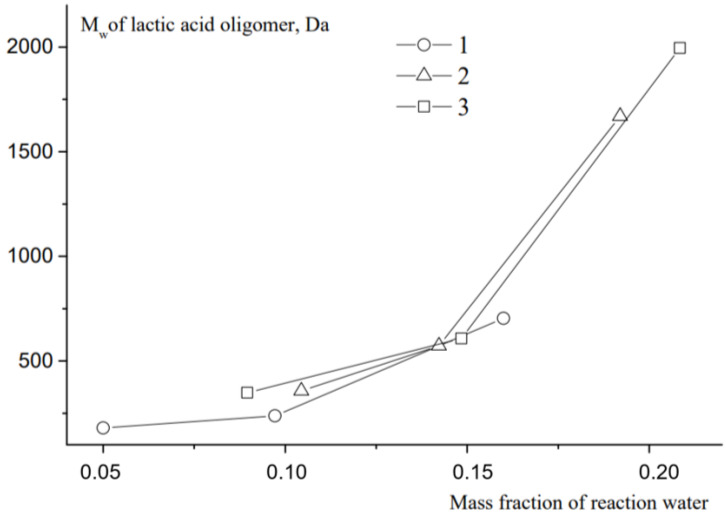
Dependence of molecular weight of oligomers on evaporated reaction water. Temperature: 25 °C. Eluent: tetrahydrofuran (THF) with a flow rate of 1.0 mL/min.

**Figure 12 polymers-16-00624-f012:**
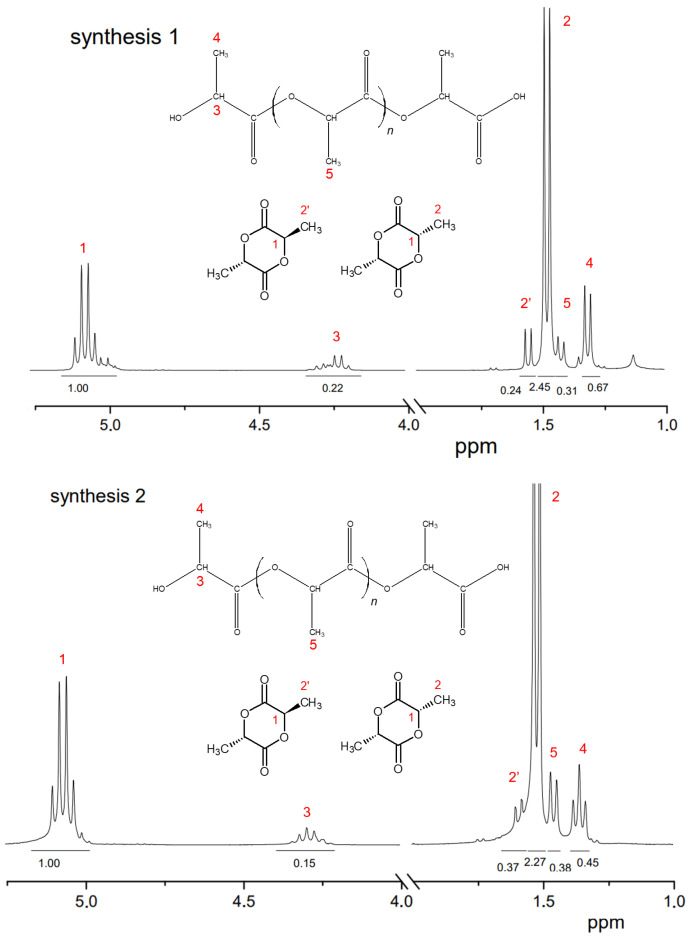
1H NMR spectra of raw lactide. Solvent: CDCl_3_ (internal standard—TMS).

**Figure 13 polymers-16-00624-f013:**
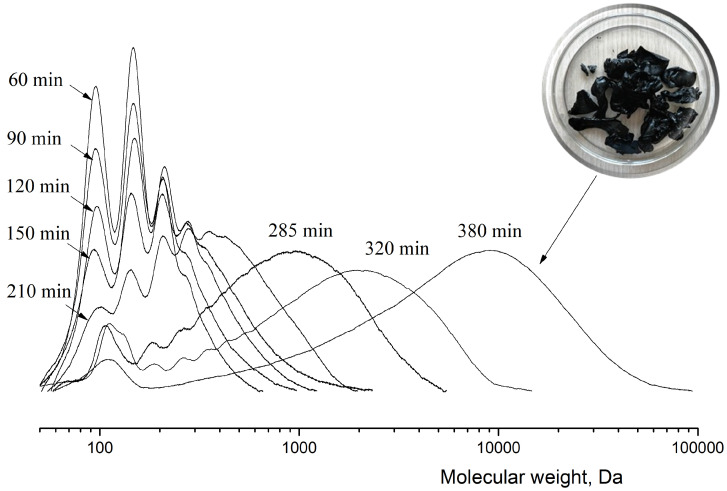
The time dependence of lactic acid oligomer molecular weight at 0.50 wt.% tin octoate catalyst content. Temperature: 25 °C. Eluent: tetrahydrofuran (THF) with a flow rate of 1.0 mL/min.

**Figure 14 polymers-16-00624-f014:**
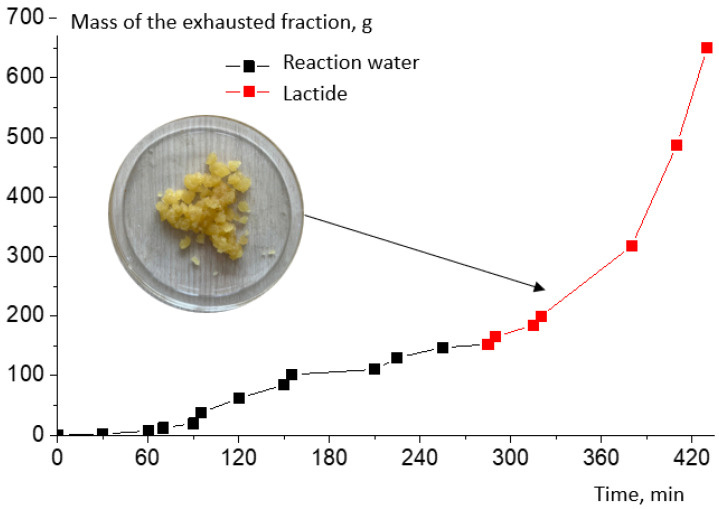
The time dependence of the evaporated fraction at 0.50 wt.% tin octoate catalyst content.

**Figure 15 polymers-16-00624-f015:**
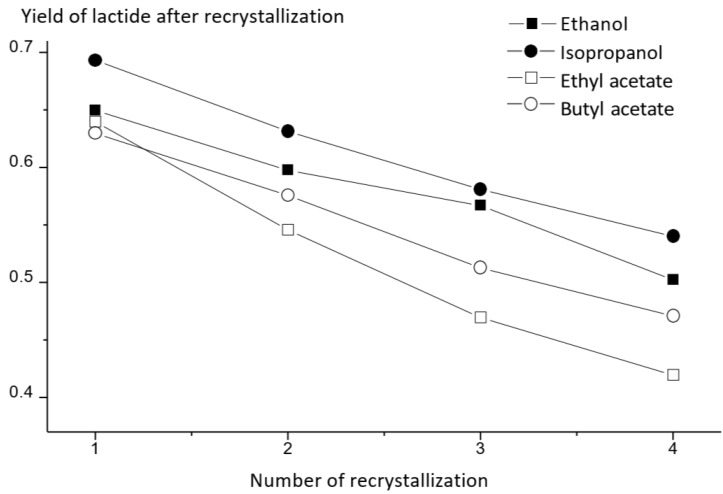
Product yields during recrystallization from different solvents.

**Figure 16 polymers-16-00624-f016:**
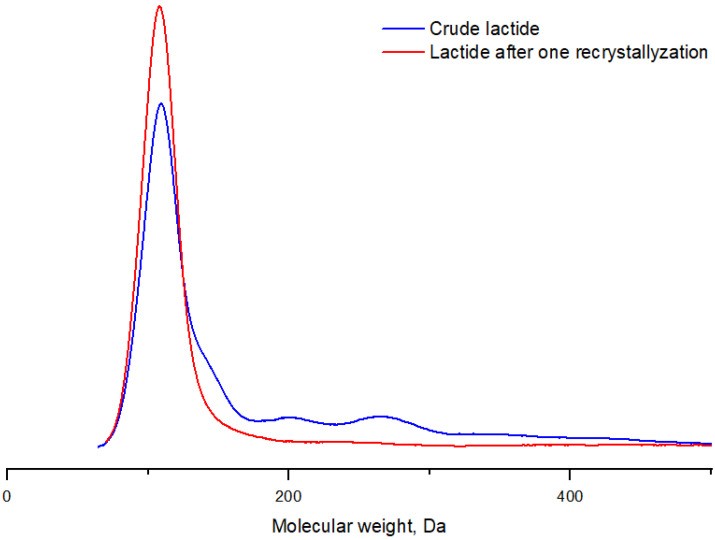
GPC curves of raw lactide before and after purification by recrystallization. Temperature: 25 °C. Eluent: tetrahydrofuran (THF) with a flow rate of 1.0 mL/min.

**Figure 17 polymers-16-00624-f017:**
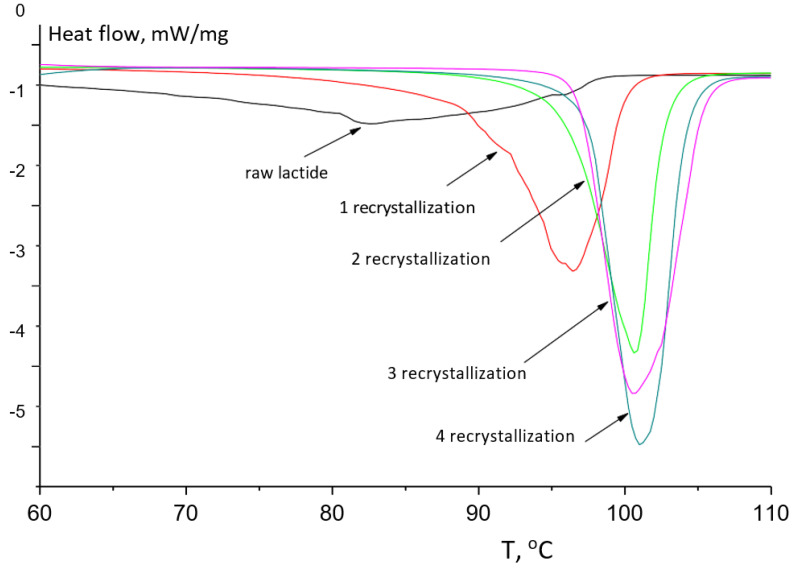
DSC curves of lactide recrystallized from butyl acetate. The inert atmosphere is argon; the flow rate is 20 mL/min; and the heating rate is 10 °C/min.

**Figure 18 polymers-16-00624-f018:**
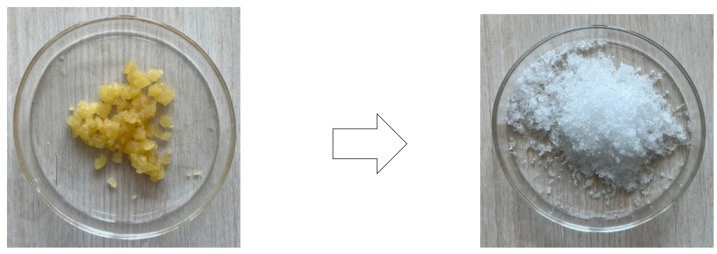
Lactide before purification and after four recrystallization iterations in butyl acetate.

**Figure 19 polymers-16-00624-f019:**
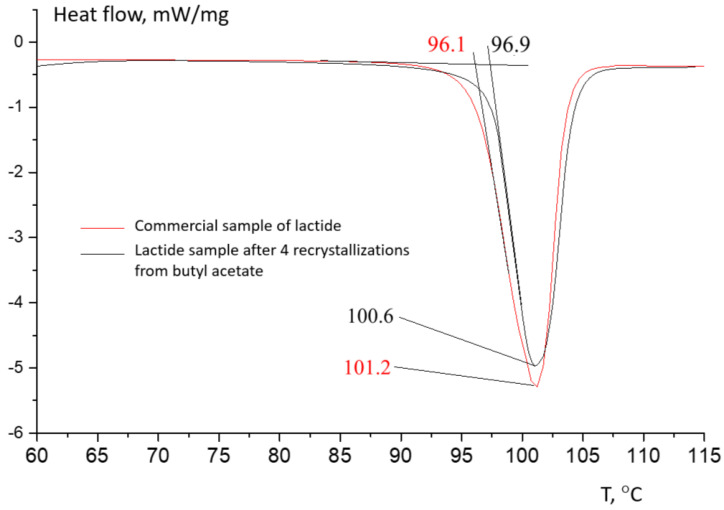
DSC curves for commercial sample of lactide and laboratory-made lactide after 4 recrystallization iterations in butyl acetate. The inert atmosphere is argon; the flow rate is 20 mL/min; and the heating rate is 10 °C/min.

**Figure 20 polymers-16-00624-f020:**
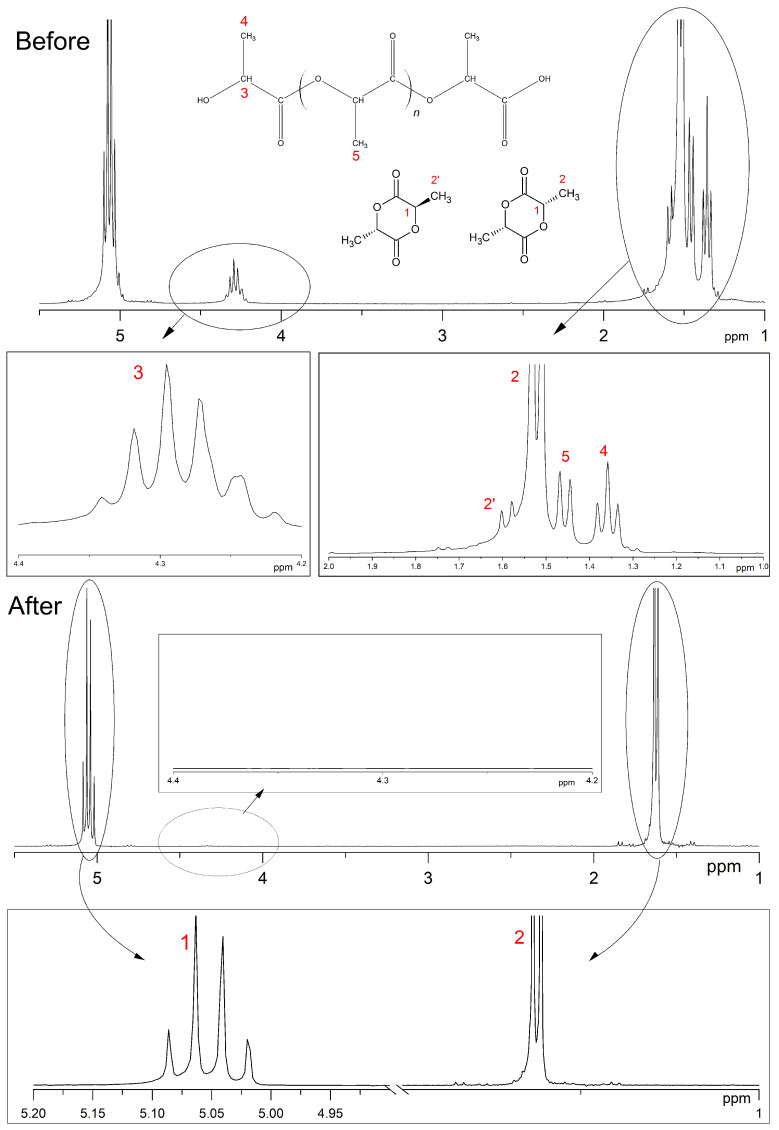
1H NMR spectra of raw lactide and lactide after the fourth recrystallization in butyl acetate. Solvent: CDCl_3_ (internal standard—TMS).

**Figure 21 polymers-16-00624-f021:**
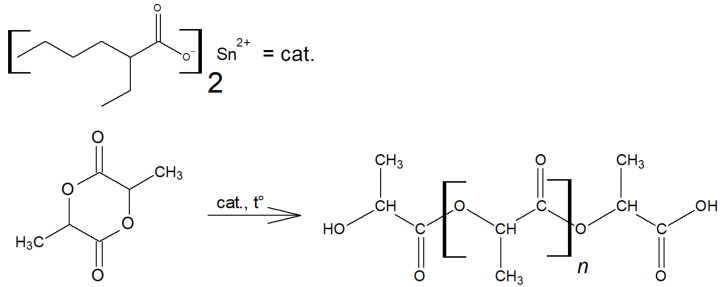
Schematic representation of the reaction for PLA production by ring-opening reaction using a tin octoate catalyst.

**Figure 22 polymers-16-00624-f022:**
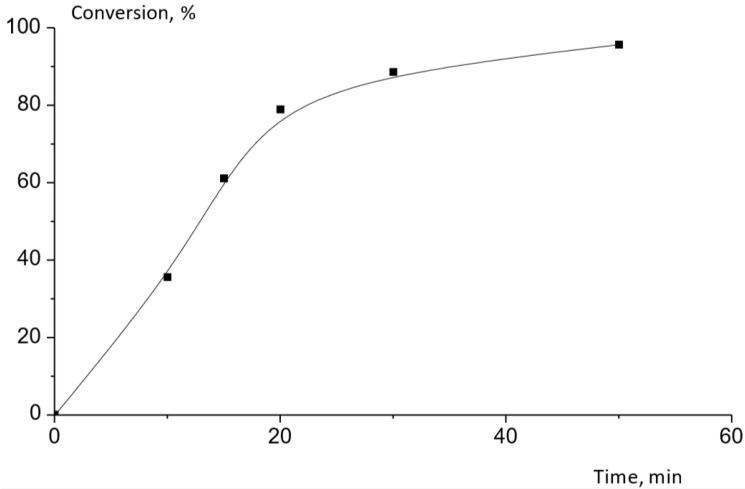
Dependence of conversion on reaction time for PLA production using tin octoate as a catalyst at 190 °C.

**Figure 23 polymers-16-00624-f023:**
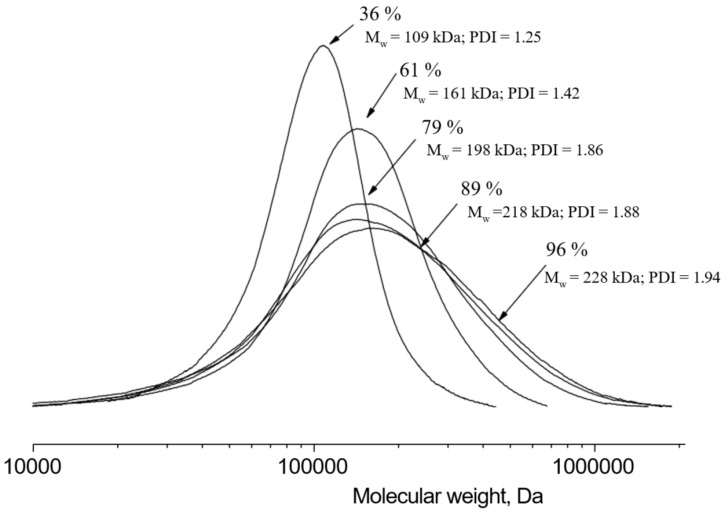
Molecular mass distribution during the synthesis of PLA using a tin octoate catalyst. Temperature: 25 °C. Eluent: tetrahydrofuran (THF) with a flow rate of 1.0 mL/min.

**Figure 24 polymers-16-00624-f024:**
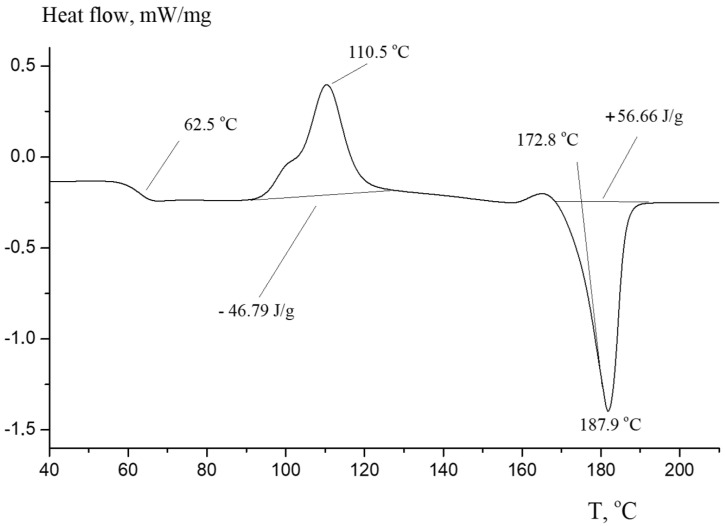
DSC curve for synthesized PLA. The inert atmosphere is argon; the flow rate is 20 mL/min; and the heating rate is 10 °C /min.

**Figure 25 polymers-16-00624-f025:**
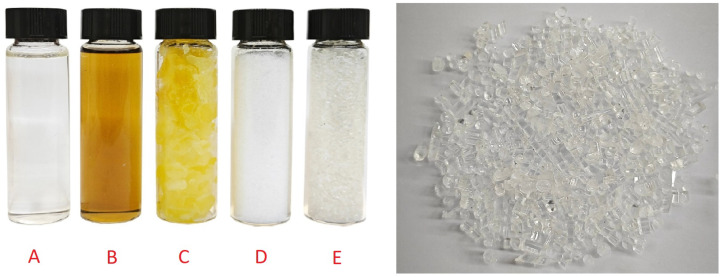
Intermediate products of polylactide synthesis: A—lactic acid; B—lactic acid oligomers; C—raw lactide; D—purified lactide; E—polylactide pellets obtained using a laboratory extruder.

**Table 1 polymers-16-00624-t001:** GPC results of lactic acid oligomers and yield of lactide.

Synthesis Number	Catalyst, % wt	Reaction Time, min	Mw	Yield of Raw Lactide, %
1	0.25	78	220	69
		100	300	
		180	860	
2	0.50	78	360	68
		125	570	
		186	1670	
3	1.00	135	350	67
		225	610	
		285	1995	
		345	7400	

**Table 2 polymers-16-00624-t002:** Initial T_melting_ values of lactide determined by the DSC method.

Iteration Number	Initial T_Melting_, °C			
	**Ethanol**	**Isopropanol**	**Ethyl Acetate**	**Butyl Acetate**
1	84.5	92.9	83.4	90.6
2	95.6	94.9	96.0	95.8
3	94.8	96.7	96.9	96.9
4	97.1	97.2	97.2	97.2

## Data Availability

Data are contained within the article.

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
