# Peer review of "Synthesis of L-Lactide from Lactic Acid and Production of PLA Pellets: Full-Cycle Laboratory-Scale Technology"

_polymers, 2024, doi:10.3390/polym16050624_

Round 1

Reviewer 1 Report

Comments and Suggestions for Authors

The review report is attached. 

Author Response

The authors are grateful to the reviewer for the attention to the manuscript and valuable comments, which allowed us to improve the quality of the presentation of the material. The responses and comments are below:

  1. The authors briefly mentions future work involving the synthesis of copolymers. Providing a brief discussion on the potential applications or advantages of these copolymers could be beneficial.

Thank you for the suggestion. We have added discussion on the potential applications (lines 366-375): The resulting molecular weight of the polymer is sufficient to synthesize copolymers of various molecular weights and structures based on purified lactide. Particularly, copolymers of L-lactide with D-lactide and/or glycolide, i.e. PLGA, could be synthesized to tailor crystallinity and mechanical properties of the materials, as well as their solubility and degradation rate. Such glycolide-enriched copolymers are widely used for fabrication of absorbable surgical sutures, while copolymers of D,L-lactide with glycolide are promising for development of novel targeted drug nanoformulations and controlled drug-release systems based on PLGA microparticles [4, 47]. Copolymerization of L-lactide with ε-caprolactone is an effective approach to tune thermal and mechanical properties of the materials, providing the way to fabricate more soft and elastic materials with an increased toughness for various biomedical and technical applications [48].

  1. The authors are encouraged to include a brief discussion or comparison with existing methods for PLA production, which could help contextualize the significance of the proposed approach.

Indeed, a review of other methods for producing PLA would have improved the article. However, there are quite a small number of articles that present the full cycle of polylactide production which starts with lactic acid. It may not make sense to describe each stage separately for comparison, since the approaches used are classic for the laboratory scale and are well described in the literature.

As for the synthesis of PLA on a pilot and industrial scale, either reactive extrusion or horizontal disk reactors paired with vertical reactors are often used [10.1002/9781119767480]. However, implementing such an approach to obtain PLA in an amount of 100-300 g would not be effective.

  1. Given the increasing importance of sustainability, discussing the environmental aspects of the proposed process, such as energy consumption or waste generation, would be valuable.

The introduction mentions the environmental friendliness, biodegradability and biocompatibility of polylactide. In our opinion, going into more detail may not be of great importance, since the article is devoted to the synthesis of PLA, and not its decomposition and analysis of the environmental impact.

  1. The figures presented in the manuscript are very well drawn except figure 7 and chemical structures in figure 12, which is a bit blurred. Please replace this figure with the one that has higher resolution

Thank you for the comment. Figures have been updated.

Reviewer 2 Report

Comments and Suggestions for Authors

The paper: “Synthesis of L-lactide from lactic acid and production of PLA

pellets: full-cycle laboratory-scale technology” presents interesting approach to problem of production of pure PLA using various methods of synthesis.

My remarks as a reviewer are:

Line 7. In abstract state the volume for your laboratory reactor to see the difference between “small scale” and "big scale”

Is there a difference in yield if scaleup procedure is used?

Line 74. Did you try or make an experiment with lactic acid produced in a fermentation of some kind?

In my opinion Figures 3, 4 and 5 are not necessary as they do not contribute to paper quality. It is just schematic presentation of your experimental setup and I do not see that anything new or innovative is implemented in your setup.

Line 84. Did you use or compare other catalysts for your oligomerization?

Line 130. Why this concentration of tin ocotate? Is this in correspondence to % in Line 86?

Author Response

The authors are grateful to the reviewer for the attention to the manuscript and valuable comments, which allowed us to improve the quality of the presentation of the material. The responses and comments are below:

  • Line 7. In abstract state the volume for your laboratory reactor to see the difference between “small scale” and "big scale”

Scaling was carried out from a 500 ml flask to a 2000 ml reactor. Volume values were added to the abstract.

  • Is there a difference in yield if scaleup procedure is used?

There were no differences in the process and reaction yield when scaling from 500 ml to 2000 ml. Further scaling was not carried out to obtain lactide with a low mesolactide content.

  • Line 74. Did you try or make an experiment with lactic acid produced in a fermentation of some kind?

No. We used commercially available concentrated technical lactic acid.

  • In my opinion Figures 3, 4 and 5 are not necessary as they do not contribute to paper quality. It is just schematic presentation of your experimental setup and I do not see that anything new or innovative is implemented in your setup.

Yes, indeed the Figures mentioned are not innovative. However, we felt it was important to indicate the setup on which the current results were obtained so that we can compare technological approaches in future work in which we plan to further scale up the process.

  • Line 84. Did you use or compare other catalysts for your oligomerization?

We used tin octoate in this work, since most studies on the synthesis of lactide report its high efficiency. This catalyst is also used in industry. However, we plan to investigate other catalysts in our future works.

  • Line 130. Why this concentration of tin ocotate? Is this in correspondence to % in Line 86?

This concentration was chosen as optimal because further increases in concentration did not lead to an increase in yield. Moreover, increasing the tin octoate content would increase the tin content of the final polymer, which could limit its applications. The line 86 mentioned by the reviewer indicates the purity of the nitrogen used to purge the mixture. That is, this % has nothing to do with the concentration of the catalyst.